# Inertial Measurement Unit-Based Frozen Shoulder Identification from Daily Shoulder Tasks Using Machine Learning Approaches

**DOI:** 10.3390/s24206656

**Published:** 2024-10-16

**Authors:** Chien-Pin Liu, Ting-Yang Lu, Hsuan-Chih Wang, Chih-Ya Chang, Chia-Yeh Hsieh, Chia-Tai Chan

**Affiliations:** 1Department of Biomedical Engineering, National Yang Ming Chiao Tung University, Taipei City 112, Taiwan; henry062439.be09@nycu.edu.tw (C.-P.L.); a0986690180.be11@nycu.edu.tw (H.-C.W.); 2Research Center for Information Technology Innovation, Academia Sinica, Taipei City 114, Taiwan; tingyang.lu.14@gmail.com; 3Department of Physical Medicine and Rehabilitation, Tri-Service General Hospital, Taipei City 114, Taiwan; gradesboy@gmail.com; 4Bachelor’s Program in Medical Informatics and Innovative Applications, Fu Jen Catholic University, New Taipei City 242, Taiwan

**Keywords:** frozen shoulder, machine learning, inertial measurement unit, identification system

## Abstract

Frozen shoulder (FS) is a common shoulder condition accompanied by shoulder pain and a loss of shoulder range of motion (ROM). The typical clinical assessment tools such as questionnaires and ROM measurement are susceptible to subjectivity and individual bias. To provide an objective evaluation for clinical assessment, this study proposes an inertial measurement unit (IMU)-based identification system to automatically identify shoulder tasks whether performed by healthy subjects or FS patients. Two groups of features (time-domain statistical features and kinematic features), seven machine learning (ML) techniques, and two deep learning (DL) models are applied in the proposed identification system. For the experiments, 24 FS patients and 20 healthy subjects were recruited to perform five daily shoulder tasks with two IMUs attached to the arm and the wrist. The results demonstrate that the proposed system using deep learning presented the best identification performance using all features. The convolutional neural network achieved the best identification accuracy of 88.26%, and the multilayer perceptron obtained the best F1 score of 89.23%. Further analysis revealed that the identification performance based on wrist features had a higher accuracy compared to that based on arm features. The system’s performance using time-domain statistical features has better discriminability in terms of identifying FS compared to using kinematic features. We demonstrate that the implementation of the IMU-based identification system using ML is feasible for FS assessment in clinical practice.

## 1. Introduction

Frozen shoulder (FS), also known as adhesive capsulitis, is an idiopathic condition associated with pain and stiffness in the shoulder joints [1]. FS is characterized by a thickened, tight glenohumeral joint capsule with adhesions obliterating the normally patulous axillary fold [2]. Most patients diagnosed with FS, especially women, are aged from 40 to 60 years [3]. The prevalence is between 2% and 5% in the general population and has a certain correlation with the conditions diabetes mellitus, shoulder injury, and Parkinson’s disease, being particularly prevalent in diabetes mellitus [4].

The symptoms of FS can be divided into three stages, including a freezing stage, a frozen stage, and a thawing stage. During the freezing and frozen stages, the patient often experiences pain and a progressive loss of active and passive range of motion (ROM) of the shoulder, which restricts patients from performing activities of daily living, including combing hair, dressing, reaching their back, doing household chores, and reaching their back pocket [5]. In the thawing stage, the symptoms of the shoulder slowly return to normal.

In clinical practice, questionnaires and measurements of shoulder ROM are important references for diagnosing FS [6,7,8,9]. Despite questionnaires being able to directly reflect the level of a patient’s symptoms, such as pain and disability, through their responses, this approach is subjective and has issues with content validity and reliability, the respondent’s interpretation, and issues of language and culture [10]. Another approach for assessing FS is to implement shoulder ROM measurements with a goniometer to present shoulder mobility in different directions. This method provides objective and quantitative assessment for treatment evaluation. However, ROM cannot directly reflect a functional capacity during daily life and work and may vary for different measurers. Moreover, the manual implementation of the measurements may lead to issues related to inter- and intra-rater reliability.

With advantages in portability, user-friendliness, and lower cost than other motion analysis systems, inertial measurement units (IMUs) have recently been widely employed to tackle the abovementioned technical issues in clinical evaluation. Previous research has applied IMU-based motion analysis systems for clinical assessment. For example, Chiang et al. [11] used wearable IMUs to record knee ROM for perioperative total knee arthroplasty (TKA). Such digitalized, continuous, and objective movement information can support surgery and manage the recovery progress of TKA patients. Palmerini et al. [12] proposed an instrumented timed up and go (iTUG) test, where an accelerometer worn on the lower back records acceleration changes during the TUG test. By extracting gait features from signals of different test components, such as sit-to-walk, gait, and turning, this study was able to quantify the motor impairment and pathological performance of Parkinson’s disease (PD) and identify healthy and early–mild PD subjects. Regarding FS assessment, several studies have focused on measuring the ROM of different movements. For instance, Ajčević et al. [13] measured the ROM in shoulder elevation and abduction by IMUs, and also provided feedback for FS patients. Furthermore, Lu et al. [14] proposed an instrumented functional assessment of the shoulder using IMUs for frozen shoulders. The results showed that kinematic features derived from an IMU signal of shoulder tasks could reflect the differences between patients with FS and healthy subjects. Additionally, they segmented a complete shoulder task into three subtasks according to different shoulder movements, which provided complementary information that corresponds to the clinical presentation of frozen shoulder.

In recent years, machine learning (ML) techniques have frequently been used to support clinical assessment, disease diagnosis, and rehabilitation monitoring. Previous studies have applied supervised learning techniques to extract the essential characteristics from complex feature patterns and to support decision making for various diseases [15]. For example, Caramia et al. classified PD patients with a set of different ML techniques based on gait analysis, and also investigated gait-related manifestations associated with the severity of the pathology [16]. Hsieh et al. developed an ML-based segmentation approach to divide different subtask segments during the TUG test for patients with perioperative TKA. Accurate subtask segments of TUG could help clinical professionals assess patient mobility and balance capability more precisely, as well as support further medical decision making [17]. Lee et al. [18] proposed a rehabilitation outcome prediction and monitoring system in stroke and traumatic brain injury survivors. Their approach, using a Gaussian process regression algorithm and wearable technologies, monitored the recovery progress and estimated the rehabilitation response for the development of a treatment management plan. This research has shown the feasibility of ML approaches in various healthcare applications.

While a great deal of research has developed ML-based FS rehabilitation monitoring systems for telehealthcare [19,20,21], there have been fewer studies that have focused on employing ML techniques to support assessment and evaluation. Batool et al. [22] first proposed an ML-based FS identification system, which extracted features based on Apley’s scratch test and resisted tests [23]; they also built classification models with logistic regression, Random Forest, and Naïve Bayes. Their system could achieve 95.1% accuracy using a logistic regression model. However, their data acquisition techniques mainly relied on manual examination and patient reports, which are time-consuming and tedious. Therefore, it is critical to develop an easy-to-use, efficient, and automatic FS identification system for FS diagnosis in medical practice.

In order to assist patients with FS to evaluate shoulder functionality by performing daily movements and to provide clinical personnel objective and quantitative information, this study aims to propose an FS identification system combining wearable sensors, ML techniques, and deep learning (DL) techniques. The system places IMUs on the arm and wrist to record movement information and extract kinematic characteristics from the collected signal. An identification model using various ML and DL techniques was developed to automatically identify the shoulder task, whether performed by healthy or injured shoulders. The main contributions of this work involve the following:We propose an IMU-based FS identification system using ML and DL techniques, which can provide portable, objective, automatic, and easily operated identification tools for clinical assessment in medical practice.This study explores the potential of various ML and DL techniques in the identification of the FS using IMU features of shoulder tasks and subtasks.This research investigates the impact of feature types, sensor placements, and dimension reduction techniques in the identification process to provide complementary information for clinical assessment.An experiment recruiting 24 FS patients and 20 healthy participants is conducted to validate the feasibility of the proposed identification system.

The rest of this paper is organized as follows. Section 2 illustrates the experiment protocol and the proposed frozen shoulder identification approach. Section 3 shows the experiment results of different ML classifiers, feature types, sensor placements, and feature selection approaches. The performance analysis, limitations, and future research are presented in Section 4. Finally, Section 5 summarizes the results of this study.

## 2. Materials and Methods

This study proposes ML-based FS identification using two IMUs attached to the wrist and the arm. Initially, the collected accelerometer and gyroscope data of the shoulder tasks are acquired and manually labeled into subtasks. Then, a series of feature extraction and selection techniques are employed to obtain diverse movement features and parameters from the raw data for identification. Lastly, classical ML and DL classifiers are applied to identify healthy and damaged shoulders using different feature subsets. The proposed framework of ML- and DL-based FS identification is shown in Figure 1.

### 2.1. Data Acquisition

#### 2.1.1. Participants

This study was approved by the institutional review board (TSGHIRB No.: A202005024) at the university hospital. Recruitment for participants took place at the rehabilitation department of the Tri-Service General Hospital. A total of 24 patients with unilateral frozen shoulder and 2 participants with bilateral frozen shoulder were recruited. The diagnosis was made by a physiatrist, and eligible patients were excluded from this study if they fell under the following criteria: full- or massive-thickness tear of the rotator cuff on magnetic resonance imaging (MRI) or ultrasonography, secondary frozen shoulder, or acute cervical radiculopathy. All patients were presented with the assessment before treatment. In total, 20 healthy subjects without a history of a shoulder condition were recruited as the control group, as was conducted in our previous study [8]. Moreover, there were 26 patients with affected shoulders and 20 healthy patients without affected shoulders on the dominant side that performed the complete experiment and were used in this study. The participants’ demographic information is summarized in Table 1.

#### 2.1.2. Experiment Protocol

In this study, we used 2 IMUs (APDM Inc., Portland, OR, USA) that incorporated a tri-axial accelerometer (range: ±16 g, resolution: 14 bits) and a tri-axial gyroscope (range: ±2000 °/s, resolution: 16 bits) to collect the data on movements. The sampling frequency of the IMUs was 128 Hz. There is supporting software (Motion studio ver1.0.0.201903301338) for all sensors to synchronize during the configuration. Each participant was then asked to place the IMUs on the wrist and the arm, as shown in Figure 2. Thus, all sensors were synchronized and calibrated before the experiment. The location of the arm IMU was approximately 5 cm above the elbow on the lateral side. The IMUs were placed on the affected side of the patient and the dominant side of the healthy subject. In the beginning, the participants were instructed to perform initial postures that involved remaining relaxed, standing with their feet at hip width, and their arms alongside the body.

They then performed 5 shoulder tasks based on the Shoulder Pain and Disability Index (SPADI) [6], including washing hair (WH), washing upper back (WUB), washing lower back (WLB), placing an object on a high shelf (POH), and removing an object from the back pocket (ROP). These tasks have been commonly considered during shoulder motion for the purpose of analyzing shoulder diseases [24,25]. During the performance, the participants carried out each shoulder task once at their own pace and in their habitual style. Note that the participants had to return to their initial postures to complete the current task before starting the next one. The IMU data were synchronized with video recording for further processing procedures.

### 2.2. Pre-Processing

The previous work showed that recording the movement performance of subtasks can reflect the differences between FS and healthy participants [8]. The raw data of each completed shoulder task were manually labeled into a complete task segment, and the complete segment was then divided into three subtask segments, as shown in Table 2. The feature extraction was applied to each segment (1 full-task segment and 3 subtask segments), and this was then concatenated into a feature vector for the training and testing data.

### 2.3. Feature Extraction and Selection

After the data pre-processing, two feature groups, including time-domain statistical and kinematic features, were applied to the signals of each complete shoulder task and subtask. These features could represent the characteristics and quality of the upper limb movement. A brief introduction is given below.

#### 2.3.1. Time-Domain Statistical Features

In total, four segments were labeled for a single shoulder task. For each segment, $\alpha_{x}$, $\alpha_{y}$, and $\alpha_{z}$ represented the *x*-, *y*-, and *z*-axial acceleration (α), respectively, and $\omega_{x}$, $\omega_{y}$, and $\omega_{z}$ represented the *x-, y-,* and *z*-axial angular velocity (ω), respectively. The Euclidean norm was then calculated using Equations (1) and (2) and denoted by $\alpha_{Norm}$ and $\omega_{Norm}$ separately. The acceleration and angular velocity on the horizontal plane, coronal plane, and sagittal plane were calculated using Equations (3)–(8) and denoted by $\alpha_{hori}$, $\omega_{hori}$, $\alpha_{coro}$, $\omega_{coro}$, $\alpha_{sagi}$, and $\omega_{sagi}$, respectively.
(1)$$\alpha_{Norm}=\sqrt{{\alpha_{x}}^{2}+{\alpha_{y}}^{2}+{\alpha_{z}}^{2}},$$
(2)$$\omega_{Norm}=\sqrt{{\omega_{x}}^{2}+{\omega_{y}}^{2}+{\omega_{z}}^{2}},$$
(3)$$\alpha_{hori}=\sqrt{{\alpha_{y}}^{2}+{\alpha_{z}}^{2}},$$
(4)$$\alpha_{coro}=\sqrt{{\alpha_{x}}^{2}+{\alpha_{z}}^{2}},$$
(5)$$\alpha_{sagi}=\sqrt{{\alpha_{x}}^{2}+{\alpha_{y}}^{2}},$$
(6)$$\omega_{hori}=\sqrt{{\omega_{y}}^{2}+{\omega_{z}}^{2}},$$
(7)$$\omega_{coro}=\sqrt{{\omega_{x}}^{2}+{\omega_{z}}^{2}},$$
(8)$$\omega_{sagi}=\sqrt{{\omega_{x}}^{2}+{\omega_{y}}^{2}}.$$

$A_{T}=\{\alpha_{x}$, $\alpha_{y}$, $\alpha_{z}$, $\alpha_{hori}$, $\alpha_{coro}$, $\alpha_{sagi}$, $\alpha_{Norm}\}$ is defined as the set of acceleration, while $W_{T}=\{\omega_{x}$, $\omega_{y}$, $\omega_{z}$, $\omega_{hori}$, $\omega_{coro}$, $\omega_{sagi}$, $\omega_{Norm}\}$ is defined as the set of angular velocity. Time-domain statistical features are commonly used for human activity analysis [26]. In this work, eight common feature types, including the mean, standard deviation, variance, maximum, minimum, range, kurtosis, and skewness, were applied to accelerometers and gyroscopes. The list of 112 features for a single IMU of a shoulder task segment is summarized in Table 3. A total of 896 (2 IMUs × 4 segments) features were extracted for a shoulder task.

#### 2.3.2. Kinematic Features

Kinematic features aim to extract movement characteristics related to smoothness, power, and speed from the IMU’s signal, which provide quantitative measurement of the movement quality and subtle information associated with clinical observation [27,28,29,30]. This work selected seven feature types to reflect the differences between patients and healthy subjects in shoulder tasks. The parameters are detailed below.

Number of mean crossing points (NMCP): NMCP measures the number of times the $\alpha_{Norm}$ value changes drastically, which suggests the change in movement direction or the fluidity of movement.Number of peaks (NP): NP measures the peaks of $\alpha_{Norm}$ during the movement segment. NP suggests the continuity and the periodicity of actions.Spectral arc length (SPARC): We applied SPARC as a negative arc length of the amplitude of $\omega_{Norm}$, which can be a candidate for the complexity of a curve’s shape. An unsmooth movement composed of multiple sub-movements leads to an increase in arc length [28], which is calculated via the following equation:(9)$$SPARC\;≜-\int_{0}^{\omega_{c}}\sqrt{{\left(\frac{1}{\omega_{c}}\right)}^{2}+{\left(\frac{d\hat{V}(\omega )}{d\omega }\right)}^{2}}d\omega ,\;d\hat{V}(\omega )≜\frac{V(\omega )}{V(0)}.$$Log dimensionless jerk (LDLJ): We applied LDLJ to $\alpha_{Norm}$. LDLJ evaluates the movement planning and control ability via the minimum jerk of the acceleration [30].
(10)$$LDLJ≜-\ln\left(\frac{{\left(t_{2}-t_{1}\right)}^{3}}{v_{peak}^{2}}\int_{t_{1}}^{t_{2}}{\left|\frac{d^{2}v}{dt^{2}}\right|}^{2}dt\right)$$

A previous study showed that LDLJ corresponds well to the reaching movement and can differentiate the movement smoothness of patients and healthy subjects.

Range of angular velocity (RAV): RAV measures the average of the range of tri-axial angular velocity, which associates with the velocity change during the task [27].Power index (PI): PI measures the average of the inner product of the range of tri-axial angular velocity and the range of tri-axial acceleration, which associates with the power control variation during the task [27].
(11)$$PI≜\;\sum_{roll,pitch,yaw}range(acceleration)\cdot range(angular\;velocity).$$Duration: Duration is determined as the interval between the start and end times of the task. This work manually labeled the duration via a recorded video. The duration was considered as starting with the wrist being left in the initial position and ending with the wrist coming back into the initial position.

Table 4 lists the extracted 13 kinematic features for each segment. A total of 52 kinematic features (13 features × 4 segments) were extracted for a shoulder task.

#### 2.3.3. Feature Standardization

After feature extraction, each feature was separately standardized by subtracting its mean (μ) and dividing by the standard deviation (σ) [31] using Equation (12):(12)$$z=\frac{x_{i}-\mu }{\sigma }.$$

Feature standardization is a common process used in ML algorithms, which make features with different scales more comparable and avoid the bias in classification caused by distributed variables.

#### 2.3.4. Principal Component Analysis (PCA)

PCA is a popular dimension reduction method that is used to retain the variance of original data by calculating principal components, which are a linear combination of the features. The computation extracts the important information from the original feature space and projects high-dimensional variables into a low-dimensional vector space [32]. After the computation, a set of new orthogonal variables called principal components (PCs) present a different degree of variation of the original data space, which can be seen as new features for the classification models detailed below. PC is defined as Equation (13):(13)$${PC}_{i}=\alpha_{1}^{i}f_{1}+\alpha_{2}^{i}f_{2}+\cdots +\alpha_{n}^{i}f_{n},$$
where *f* and *α* are the original feature and its weight, i suggests the order of principal components, and *n* represents the number of features. This work tested the system performance by employing PCA to the features with a threshold of 95% and 99% on the retained variance.

### 2.4. Conventional ML-Based FS Identification

After data pre-processing, two feature groups including time-domain statistical and kinematic features were applied to the signals of each complete shoulder task and subtask. These features could represent characteristics and quality of upper limb movement. In this study, we applied different ML-based classifiers including support vector machine (SVM), K-nearest neighbors (KNN), decision tree (DT), random forest (RF), naïve Bayes (NB), adaptive boosting (AdaBoost), and extreme gradient boosting (XGBoost). We examined the commonly used radial basis function (RBF) as the kernel function for the SVM classifier. For the KNN classifier, a range of k from 1 to 30 was tested, and the k value with the best accuracy is presented in the experiment results.

### 2.5. DL-Based FS Identification

#### 2.5.1. Multilayer Perceptron (MLP)

Table 5 lists the employed MLP architecture. Additionally, we explored the MLP model with hyperparameters such as the layers of a fully connected layer, the dropout layer, hidden units, the learning rate, and the dropout rate, as listed in Table 6.

#### 2.5.2. Convolutional Neural Network (CNN)

We determined a convolution block containing a convolution layer, batch normalization layer, and max-pooling layer, and we examined the identification performance with different numbers of blocks and parameters. Table 5 lists the model architecture of CNN and Table 6 presents the examined hyperparameters in CNN.

### 2.6. Implementation Details

To compare the identification performance of different feature groups, we considered the feature sequence of a single instance as the input, and the input size of the 1D convolution model varied from the feature number of each feature group. The activation function of the fully connected layer and the convolution layer was ReLu and the output layer was Softmax. The optimizer was Adam. Each model was trained for a maximum 100 epochs and was stopped early for 10 iterations with a minimum loss decrease of 0.01. The model was trained and validated on a PC that had an Intel i7-9700 3.00 GHz CPU (Intel, Santa Clara, CA, USA), 32 GB RAM, and an Nvidia GTX 1650 GPU (Nvidia, Santa Clara, CA, USA).

### 2.7. Performance Evaluation

This work used leave-one-subject-out cross-validation (LOSOCV) to validate the proposed FS identification. Initially, the data were spilt into *n* subsets based on the number of subjects. LOSOCV determines the five tasks performed by one subject as the testing data, while the tasks performed by other subjects are used as the training data. Then, it iterated *n* times until all of the subjects had been used as testing data. Finally, the testing performance of *n* subsets was averaged and outputted as the final testing performance.

In the performance evaluation of FS identification, true positive (*TP*) was defined as the classifier correctly identifying an FS patient performing the shoulder task, whereas true negative (*TN*) was defined as the classifier correctly identifying a healthy subject performing the shoulder task. False positive (*FP*) was defined as the classifier misidentifying an FS patient performing the shoulder task, whereas false negative (*FN*) was defined as the classifier misidentifying a healthy subject performing the shoulder task. Four performance metrics were used to evaluate the model performance, including accuracy, recall, precision, and F1 score. The accuracy measures the correct subject prediction of tasks to the total number of sample tasks. Recall is the capability to detect the patient with a frozen shoulder, and precision is the quality to predict the exact patient. F1 score is the weighted average considering both recall and precision. The evaluation metrics were computed with Equations (14)–(17):(14)$$Accuracy=\frac{TP+TN}{TP+FP+TN+FN},$$
(15)$$Recall=\frac{TP}{TP+FN},$$
(16)$$Precision=\frac{TP}{TP+FP},$$
(17)$$F1-score=\frac{2\times Recall\times Precision}{Recall+Precision}$$

## 3. Results

Table 7 presents the experimental results of the shoulder task performer identifications that used all the features and sensors for the ML and DL techniques. In general, the classification accuracy and F1 score ranged from 72.17% to 88.26% and 75.19% to 89.23%, respectively. CNN-based identification demonstrates the best identification accuracy (88.26%) and precision (94.78%), and MLP-based identification shows the best F1 score (89.23%) and recall (89.23%). SVM achieved the best identification accuracy of 83.91% and an F1 score of 85.49% with ML-based identification, which was roughly 3% lower than those with the DL-based classifiers. Additionally, DT obtained the worst accuracy of 72.17% among the classifiers. 

As shown in Figure 3, the identification that used typical time-domain statistical features presented better discriminative ability than that which used kinematic features in most of the classifiers. The improvement ranged from 2.17% to 9.13% and 1.89% to 10.09% in accuracy and F1 score, respectively. MLP-based identification that used statistical features had the best identification accuracy of 86.52% and an F1 score of 87.94%; the CNN-based classifier demonstrated similar identification performance.

Figure 4 displays the identification performance using features extracted from different sensor placements. The results demonstrate that most classifiers achieved better identification accuracy and F1 score with wrist placement features than with arm features except MLP. The improvement in accuracy ranged from 0.44% to 12.59% in most classifiers, while the few remained the same or slightly decreased. Among all the classifiers, the DT-based classifier that used wrist features significantly improved the accuracy by 12.59% and the F1 score by 10.41%. CNN-based identification that used wrist features presented the best accuracy (84.78%), recall (89.23%), and F1 score (86.89%).

Figure 5 shows the identification results for two feature subsets, which represented 95% and 99% retained variance of all the features after PCA was applied. The feature subsets of 95% and 99% variance were composed of 4 and 11 principal components, respectively. These two subsets produced a similar accuracy and F1 score to the evaluated machine learning method. The MLP-based classifier, which achieved an accuracy of 79.57% and an F1 score of 81.99% with a feature subset of 95% retained variance, performed better than the other techniques.

## 4. Discussion

This study presented an IMU-based FS discrimination system that uses ML and DL techniques to identify whether shoulder tasks are performed by FS patients or healthy subjects. We explored the impact of the sensor placements, feature groups, and feature reduction technique on the identification performance. Our results validate the effectiveness of the DL techniques to identify FS and show the feasibility of implementing the proposed identification system in clinical practice.

Previous studies using IMU-based shoulder motion analysis have focused on evaluating objective kinematic parameters when comparing patients and healthy subjects [27,33], and they have monitored the improvement of shoulder function before and after rotator cuff surgery [25] or rehabilitation for hemiparesis [34]. However, few studies have applied kinematic features and ML techniques to investigate the shoulder function of FS patients [13,24]. To the best of our knowledge, this is the first study to use DL techniques and IMU features in the discrimination of FS patients from healthy subjects.

Most studies have placed multiple sensors on different parts of body [35], such as the scapula, the sternum, the humerus, and the wrist [13,24], to assess shoulder function. However, few studies have explored the feasibility of assessment systems using a single unit [25,27,36]. The number and placement of sensors would directly influence user experience, usage intension, and the system cost. The results show that the proposed identification system when using CNN and the wrist sensor achieved the best accuracy (84.78%), which was slightly lower than the best performance (88.26%) when using CNN and both IMUs on the arm and the wrist. In addition, it demonstrated that using a DL-based classifier and a single IMU is feasible for achieving reliable identification performance that outperforms typical ML-based classifiers with both IMUs. Reducing the number of IMUs would make the system more friendly, easy-to-use, and useful in clinical practice.

Furthermore, we compared the identification performance using wrist features and arm features to discover suitable placements for FS assessment, as shown in Figure 4. Most classifiers using wrist features achieved better performance than those using arms. In particular, the differences between the wrist and the arm of DT and XGBoost were 7% and 12%, respectively. The results show that the features extracted from the wrist sensor were more effective in discriminating FS patients from healthy subjects compared to those obtained from the arm sensor. It revealed that FS has a great impact on the movement performance of the distal part, which makes a clear distinction in the movement patterns of wrist sensors between healthy subjects and FS patients. Similar to previous research, Mackenzie et al. [37] demonstrated that wrist-mounted accelerometers could count limb activity and support assessments before and after the treatment of FS patients. This study provides insight into the selection of sensor placement for the analysis of FS motion characteristics.

Although PCA can effectively reduce the feature dimension by composing several PCs to represent difference retained variance, the discriminant power was notably lower than that using all features. Similar results were also presented in [16], where the group of PCAs on the overall features obtained lower classification accuracy than the group of randomly selected features. Further research should explore other efficient feature selection methods to extract critical features for reductions in feature dimensions.

Previous studies have analyzed shoulder motion using kinematic features derived from IMUs instead of using statistical time-domain features to describe motion difference and to differentiate patients and healthy subjects. For example, Coley et al. [27] first proposed using a P score calculated from the humerus acceleration and angular velocity to present the difference between patients with shoulder pathology and healthy subjects for nine daily activities based on the Simple Shoulder Test. Bavan et al. [36] showed that IMU features, including smoothness, speed, and power, demonstrate effective psychometric properties in back to bulb assessments for patients with subacromial shoulder pain, and they also demonstrated that it has diagnostic power in individual arm scores and asymmetry scores. However, our analysis revealed that statistical time-domain features have better discriminant capability than kinematic features. Such results show that further exploration of more effective kinematic features for FS movement analysis is required to support assessment and diagnosis ability.

Figure 6 shows the example signals of the WLB performed by a healthy subject and a patient with FS. It shows that healthy subjects can perform natural and fluent movements and that the motion signal has notable changes in acceleration and angular velocity. By contrast, the signals collected from the patient with FS are relatively flat, which reveals the patient has difficulty in performing WLB tasks. Obviously, the FS symptoms, including pain and limited ROM, have a great impact on the performance of shoulder tasks in patients with FS.

There are several limitations to this study. The first is the limited number of participants involved in this work. More age-matched control subjects will need to be recruited to validate the proposed discrimination system, which may improve the generalization of the proposed assessment and discover essential movement features that could be used to discriminate FS patients from healthy subjects. Another one is that manual feature extraction has limited identification performance with basic DL classifiers. Hand-crafted features help observers understand and interpret the relationship between features and movement differences; however, they might not fully represent the original motion signals. Hence, more representative features need to be explored.

## 5. Conclusions

This study validated the ability of an IMU-based discrimination system using ML and DL techniques to discriminate patients with a frozen shoulder from healthy subjects. We investigated the impact of IMU placement and feature type on the identification performance. The analysis results show that the features extracted from the wrist sensor were more effective in discriminating FS patients from healthy subjects compared to those obtained from the arm sensor. Additionally, the IMU-based discrimination system that used statistical time-domain features had better discriminant capability than kinematic features, which showed that the further exploration of effective kinematic features is required for IMU-based FS assessments. In future work, the proposed IMU-based identification approach will be used to analyze the severity and affected areas of FS as it has the potential to support precise diagnosis and treatment planning in clinical practice.

## Figures and Tables

**Figure 1 sensors-24-06656-f001:**
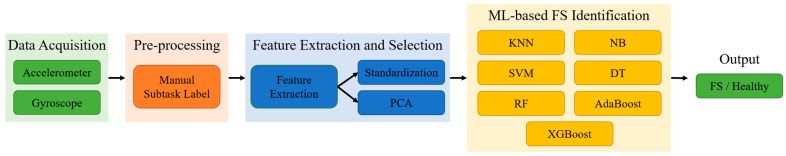
The proposed framework for ML- and DL-based frozen shoulder identification systems.

**Figure 2 sensors-24-06656-f002:**
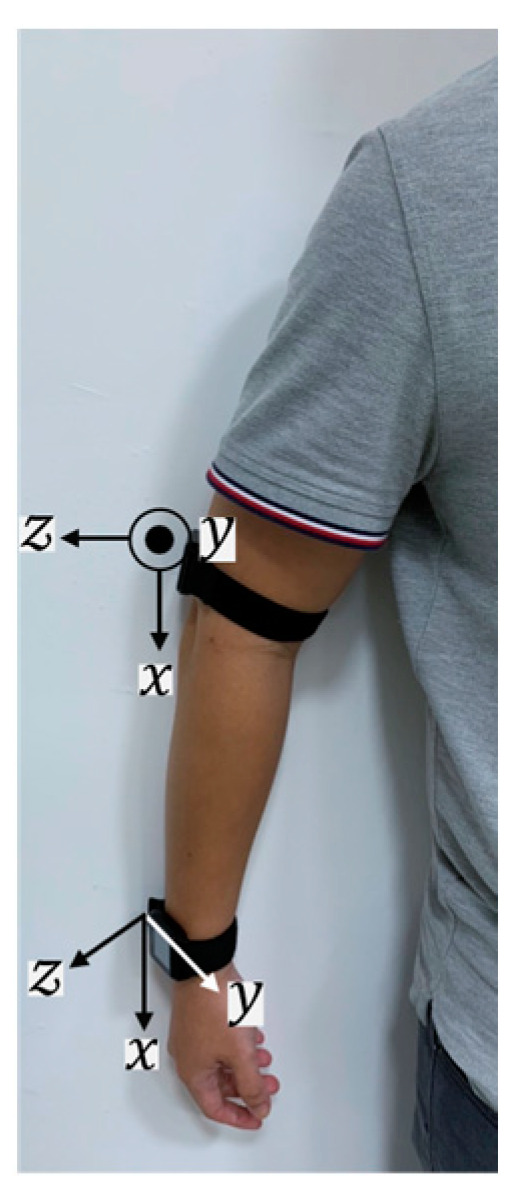
The IMU placement and axes on the right side of the upper limb.

**Figure 3 sensors-24-06656-f003:**
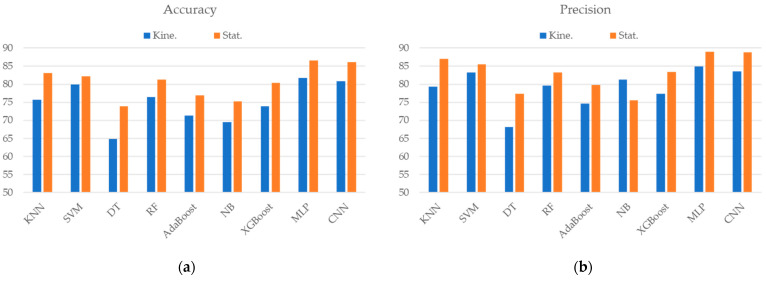
The classification performance of the (**a**) accuracy, (**b**) precision, (**c**) recall, and (**d**) F1 score when using different ML/DL techniques and types of features. Kine. and Stat. represent the kinematic features and time-domain statistical features, respectively.

**Figure 4 sensors-24-06656-f004:**
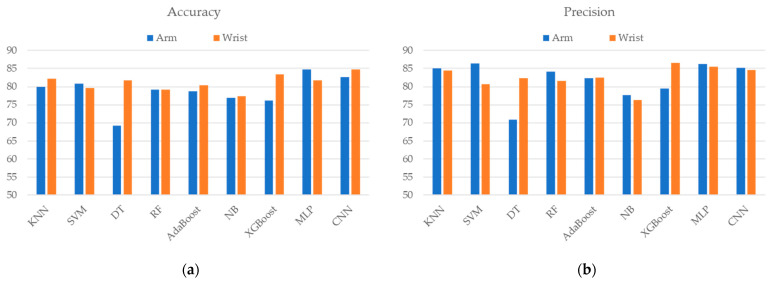
The classification performances of the (**a**) accuracy, (**b**) precision, (**c**) recall, and (**d**) F1 score when using different ML/DL techniques and placement of sensors.

**Figure 5 sensors-24-06656-f005:**
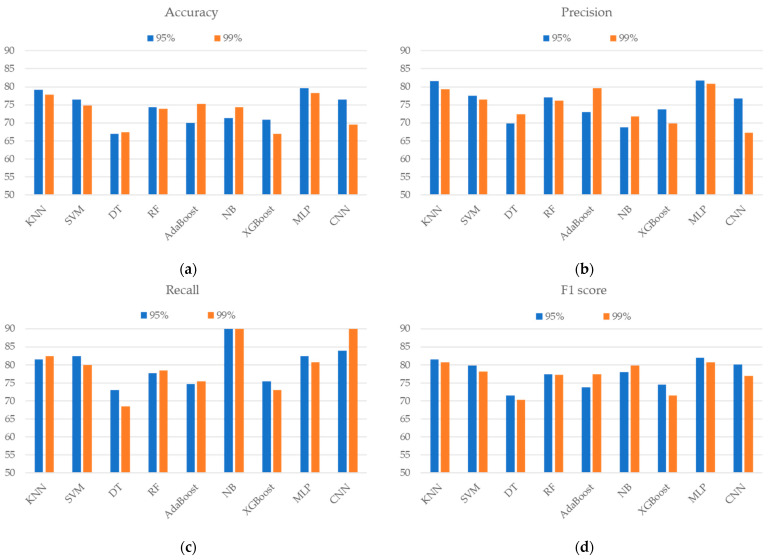
The classification performance of the (**a**) accuracy, (**b**) precision, (**c**) recall, and (**d**) F1 score when using different ML/DL techniques and PCA variance.

**Figure 6 sensors-24-06656-f006:**
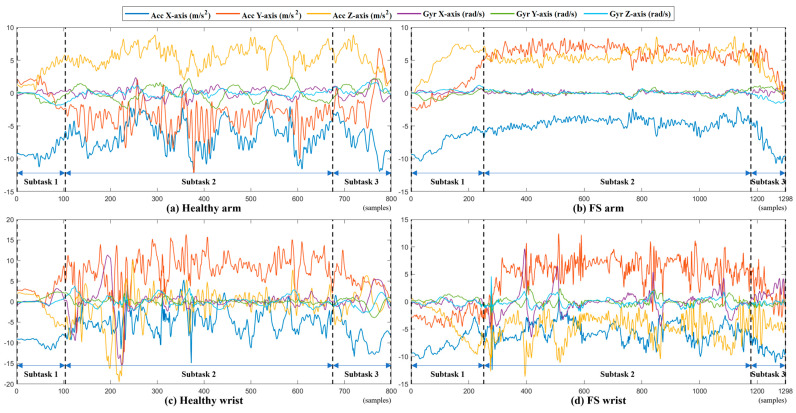
A complete task and three subtask segment signals of a healthy subject and an FS patient performing WLB. (Acc: accelerometer, Gyr: gyroscope.) (**a**) Healthy arm; (**b**) FS arm; (**c**) healthy wrist; and (**d**) FS wrist.

**Table 1 sensors-24-06656-t001:** The demographic information of the participants.

	Patients	Healthy
Sample shoulder	26	20
Gender	11 male, 13 female	10 male, 10 female
Shoulder side	13 right, 13 left	17 right, 3 left
Mean age (SD, years)	56.8 (10.6)	24.6 (3.8)
Mean height (SD, m)	1.636 (0.078)	1.686 (0.067)
Mean weight (SD, kg)	61 (12.1)	68 (15.3)

**Table 2 sensors-24-06656-t002:** Shoulder task description.

Task	Subtask	Explanation
Washing Hair (WH)	1	Lift hands toward the top of the head.
	2	Wash hair for a few seconds.
	3	Put down hands and return to the initial position.
Washing Upper Back (WUB)	1	Lift hand toward the neck.
	2	Washing the upper back for a few seconds, including shoulders on both sides and the neck.
	3	Put down the hand and return to the initial position.
Washing Lower Back (WLB)	1	Rotate the hand toward the back.
	2	Wash the lower back for a few seconds, involving the area between the shoulder blade and the waist.
	3	Put down the hand, and return to the start position.
Placing an Object on a High Shelf (POH)	1	Hold a smartphone using the painful/dominant hand in the initial position, then lift the hand to the height approximately above the head.
	2	Holding the phone in the air for a few seconds.
	3	Put down the hand and return to the initial position.
Removing an Object from the Back Pocket (ROP)	1	Hold a smartphone the painful/dominant hand in the initial position, then rotate the hand toward the back pocket.
	2	Put the phone into the pocket, then take it out.
	3	Put down the hand and return to the initial position.

**Table 3 sensors-24-06656-t003:** The feature vectors for the time-domain statistical features.

Feature Vector $F=(f_{1}^{t},f_{2}^{t},\cdots ,f_{112}^{t})$	Feature Description
$f_{1}^{t}\sim f_{14}^{t}$	Mean of $A_{T}$, $W_{T}$
$f_{15}^{t}\sim f_{28}^{t}$	Standard deviation of $A_{T}$, $W_{T}$
$f_{29}^{t}\sim f_{42}^{t}$	Variance of $A_{T}$, $W_{T}$
$f_{43}^{t}\sim f_{56}^{t}$	Maximum of $A_{T}$, $W_{T}$
$f_{57}^{t}\sim f_{70}^{t}$	Minimum of $A_{T}$, $W_{T}$
$f_{71}^{t}\sim f_{84}^{t}$	Range of $A_{T}$, $W_{T}$
$f_{85}^{t}\sim f_{98}^{t}$	Kurtosis of $A_{T}$, $W_{T}$
$f_{99}^{t}\sim f_{112}^{t}$	Skewness of $A_{T}$, $W_{T}$

**Table 4 sensors-24-06656-t004:** The feature vectors for kinematic features.

Feature Vector $F=(f_{1}^{t},f_{2}^{t},\cdots ,f_{13}^{t})$	Feature Description
$f_{1}^{t}\sim f_{2}^{t}$	NMCP of the arm and the wrist
$f_{3}^{t}\sim f_{4}^{t}$	NP of the arm and the wrist
$f_{5}^{t}\sim f_{6}^{t}$	SPARC of the arm and the wrist
$f_{7}^{t}\sim f_{8}^{t}$	LDLJ of the arm and the wrist
$f_{9}^{t}\sim f_{10}^{t}$	RAV of the arm and the wrist
$f_{11}^{t}\sim f_{12}^{t}$	PI of the arm and the wrist
$f_{13}^{t}$	Duration of the wrist

**Table 5 sensors-24-06656-t005:** MLP and CNN model architecture.

MLP	CNN
Input layer	
Fully Connected Layer 1	Convolution Layer 1
Dropout Layer 1	Batch Normalization Layer 1
Fully Connected Layer 2	Max-pooling Layer 1
Dropout Layer 2	Convolution Layer 2
Fully Connected Layer 3	Batch Normalization Layer 2
Dropout Layer 3	Max-pooling Layer 2
-	Dropout Layer
-	Fully Connected Layer
Output layer	

**Table 6 sensors-24-06656-t006:** MLP and CNN hyperparameter values.

Hyperparameter	MLP	CNN
Number of fully connected layers	1, 2, 3	1
Number of convolution blocks	-	1, 2
Kernel size	-	3, 5, 7
Filter number	-	16, 32, 64
Stride	-	1
Pooling size	-	2
Hidden units	16, 32, 64	
Dropout rate	0.1, 0.3, 0.5	
Learning rate	0.001, 0.0005, 0.0001	

**Table 7 sensors-24-06656-t007:** The identification results when using all features and sensors.

	Accuracy (%)	Precision (%)	Recall (%)	F1 Score (%)
KNN	82.61	83.09	86.92	84.96
SVM	83.91	87.2	83.85	85.49
DT	72.17	75.78	74.62	75.19
RF	78.70	80.45	82.31	81.37
AdaBoost	75.22	77.44	79.23	78.33
NB	76.09	76.22	83.85	79.85
XGBoost	80.43	84.00	80.77	82.35
MLP	87.83	89.23	**89.23**	**89.23**
CNN	**88.26**	**94.78**	83.85	88.98

Each bold result represents the best performance in the accuracy, precision, recall, and F1 score columns.

## Data Availability

The data are not publicly available due to privacy or ethical restrictions.

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
