# Peer review of "Inertial Measurement Unit-Based Frozen Shoulder Identification from Daily Shoulder Tasks Using Machine Learning Approaches"

_sensors, 2024, doi:10.3390/s24206656_

Round 1

Reviewer 1 Report

Comments and Suggestions for Authors

The paper is very interessting and well-writen.

I have some concerns and comments on the paper, which are listed below:

Section 2.1.2:
- The authors should provide information about the time synchronisation between the data of the two IMUs
- Location of the second IMU ("arm") should be described more precicely.

Section 2.3.1:
- The time-domain features are calculated over a certain time window (e.g. the mean of alpha_x, etc.). It is unclear if there is a time-window or the complete segment is used. If the latter is the case, then it is unclear if the task segment or the subtask segment is used.
- same holds for the kinematic features in section 2.3.2.
- it is unclear if the subtask segments or the whole sequence is fed into the classifiers. Authors should give more information about this (especially why the subtask segment labels are needed).

Section 2.4.6:
- typo in line 295 (avoid)

Section 2.5.1:
- as raw data is fed into MLP and CNN, the structure of this raw data is crucual. The authors should provide information about the input size and the input channel arrangement (measurement-channel order, IMU order, etc.).

Section 2.7:
- performance of the LOSOCV runs is averaged. By this, outliers in the results may be hidden by the overall average performance. In the medical use case, it may be of high interesst to know also the worst performance of all trials. This provides insights into the limits of the procedure, if the algorithms fails for single subjects. For example, quantiles and median or max/min performance could be provided in addition to the avaor boxplots of all runs.

Section 4:
- The results in this study are not compared to other results in literature for the same problem. As the authors state in line 390ff, this study first implies DL/ML techniques for the classification tasks. Therefore, the results should somehow be compared to conventional approaches (without ML) to argue why ML/DL should be used.
(Sidenote: especially DL has drawbacks, as for example the blackbox character, which is not interpretable. )

Comments on the Quality of English Language

only some typos. English language is easy to understand.

Author Response

We would like to thank you for your detailed and valuable comments. Our detailed revision note is in the attachment.

Reviewer 2 Report

Comments and Suggestions for Authors

1. The abstract section should be modified. The background introduction is excessive. Please remove the introduction part between lines 15 and 18 to make it concise. 

2. In Table 2, the last step for each entry is the same. Please revise the table to reflect this description. In Table 3, '𝛼𝑥, 𝛼𝑦, 𝛼𝑧, 𝛼ℎ𝑜𝑟𝑖, 𝛼𝑐𝑜𝑟𝑜, 𝛼𝑠𝑎𝑔𝑖 , 𝛼𝑁𝑜𝑟𝑚' and '𝜔𝑥, 𝜔𝑦, 𝜔𝑧, 𝜔ℎ𝑜𝑟𝑖 , 𝜔𝑐𝑜𝑟𝑜, 𝜔𝑠𝑎𝑔𝑖, 𝜔𝑁𝑜𝑟𝑚' appear repeatedly. Please revise the table to make it more concise.

3. There are some confusions or errors. Between lines 348 and 350, “CNN based identification demonstrates the best identification accuracy (88.26%) and F1-score (88.98%), and MLP-based identification shows the best F1-score (89.23%). ”. Please note that for the CNN-based method, besides identification accuracy, the best result is Precision (94.78%), rather than F1-score. Also, please explain why the Precision, Recall, and F1-score for the MLP-based method are all 89.23%.

4. For the introduction of methods, including subsection 2.4 from lines 248 to 298 and subsection 2.5 from lines 299 to 319, please move the relevant introduction content to the Introduction section, leaving only the implementation details in this section.

5. The results presented between Table 7 and Table 10 are difficult to read. Please also represent these tables in a plot format.

6. Please remove empty Section 6, "Patents".

Comments on the Quality of English Language

The quality of the English is good, no major issues were identified.

Author Response

(The authors gave the same response as above.)

Round 2

Reviewer 1 Report

Comments and Suggestions for Authors

Thanks for the open discussion and for improving your paper.